# Unlocking epitope similarity: A comparative analysis of the American manatee (*Trichechus manatus*) IgA and human IgA through an immuno-informatics approach

**Mariapaula Díaz-Yayguaje[1], Susana Caballero-Gaitan[2], Augusto Valderrama-Aguirre[1]***

**1** Departamento de Ciencias Biológicas, Grupo Instituto de Investigaciones Biomédicas, Universidad de Los Andes, Bogotá D.C., Colombia, **2** Departamento de Ciencias Biológicas, Laboratorio de Ecología Molecular de Vertebrados Acuáticos, Universidad de Los Andes, Bogotá D.C., Colombia

\* a.valderramaa@unaindes.edu.co

## Abstract

The American manatee (Trichechus manatus), experiencing population declines due to various threats, is the focus of conservation efforts that include the capture, rehabilitation, and release of orphaned calves when their mothers are unable to care for them. These efforts are compromised by the use of commercially available milk substitutes that lack essential components found in natural manatee breast milk, particularly immunoglobulin A (IgA). IgA plays a crucial role in nurturing the immune mucosal system and fostering a healthy microbiota. However, research on IgA in non-maternally fed manatees is limited due to the lack of species-specific reagents. To address this gap, our study employs immuno-informatics analysis to compare IgA sequences from manatees with those from other species, aiming to explore epitope similarity and sharing. We compared the protein sequence of manatee IgA with available IgA sequences, assessing similarity at the sequence, 3D structures, and epitope levels. Our findings reveal that human IgA exhibits the highest similarity in terms of sequence and 3D structure. Additionally, epitope analysis shows high conservation, identity, and similarity of predicted epitopes compared to human IgA. Future studies should focus on functional analysis using human IgA polyclonal reagents to detect manatee IgA in breast milk. Our findings highlight the potential of comparative analysis in advancing the understanding of immunology in non-human animals and overcoming challenges associated with the scarcity of species-specific reagents.

## Introduction

The American manatee (*Trichechus manatus*) is an aquatic mammal widely distributed from southern United States to Northern South America [1]. Currently, the manatee's conservation status is classified as vulnerable across its entire geographical range by the International Union for Conservation of Nature (IUCN) [2] Red List and it is found in the Convention on International Trade in Endangered Species of Wild Fauna and Flora (CITES) Appendix I [3]. This

**Data Availability Statement:** All relevant data are within the paper and its Supporting Information files.

**Funding:** This work was supported by a FAPA project (PVI0122029) from Universidad de Los Andes, granted to A.V.A. The funders had no role in study design, data collection and analysis, decision to publish, or preparation of the manuscript.

**Competing interests:** The authors have declared that no competing interests exist.

status is influenced by various factors. Firstly, manatees exhibit a slow population growth due to low birth rates, averaging around 1 calf every 2–3 years per female, and prolonged periods of parental care [4]. Secondly, they face natural challenges such as cold-water temperatures, climate-related events like hurricanes, red tide effects, and susceptibility to various pathogens, all contributing to decreased life expectancy. Finally, anthropogenic factors pose significant threats including overexploitation, hunting, habitat loss, chemical pollution, entanglement in fishing nets, and collisions with boats, with the latter being the most impactful [2,5].

The survival of orphaned manatee calves' centers on conservation initiatives that encompass their capture, rehabilitation, and eventual return to the wild [6]. Unfortunately, these substitutes either lack essential components found in natural manatee milk or contain components such as lactose in varying concentrations, leading to gastrointestinal problems [6,7]. Consequently, orphaned manatees present a significant conservation challenge as they often struggle with recovery and successful release. This is largely attributed to the inadequate nutrition provided by these substitutes, which can lead to various health issues including inflammation, gastrointestinal problems, and susceptibility to infections [7]. Despite the recognition of the importance of proper nutrition for orphaned calves, existing milk substitutes fail to encompass all the vital components required for their optimal growth and development.

Breast milk plays an important role in nurturing a robust immune mucosal system, boasting a significant amount of components including enzymes, cytokines, complement factors, and notably, immunoglobulin A (IgA). Among these, IgA stands out as the predominant immunoglobulin in mucosal tissues and secretions [8,9]. Its significance lies in its pivotal role in fortifying early and mucosal protection. IgA achieves this by preventing microbial invasion into tissues, immobilizing pathogens, and neutralizing toxins and virulence factors within the offspring's gastrointestinal tract [8]. Moreover, it acts as a shield against digestive and respiratory illnesses until the offspring's immune system matures to assume these functions independently [9,10]. In the era of metagenomics, the pivotal role of lactation, with its rich array of components including IgA, in fostering a healthy intestinal microbiota is widely acknowledged [11].

Despite the well-established importance of IgA in shaping the mucosal immune system and fostering a healthy microbiota in mammals, research on this immunoglobulin's role in non-maternally fed manatees remains scarce. The lack of species-specific reagents and reports on cross-reactivity with reagents from other species has hindered studies on IgA in manatees [12]. In response to this challenge, we conducted an immuno-informatics analysis to compare IgA sequences from manatees with those from other species, aiming to explore epitope similarity and sharing.

## Methods

### Sequence retrieval and analysis of primary structure

We retrieved manatee IgA protein sequences from the NCBI Proteins database and conducted multiple rounds of BLASTp analyses [13] only the first 100 results of each BLAST were considered. In the first round, we used all complete sequences of manatee IgA and filtered out hypothetical proteins, unnamed protein products, and variable regions from the BLASTp hits. For the second round, we excluded the human taxon (Taxon ID: 9606) to enrich hits over more species and included a dog (*Canis lupus familiaris*) IgA protein sequence with a high identity percentage [14]. A third round of BLASTp was performed in UniProt [15], using the longest manatee IgA protein sequence, and only hits corresponding to IgA were retrieved. The protein sequences were downloaded in FASTA format and aligned using MUSCLE in MEGA11 [16] with default parameters. Neighbor Joining trees were then constructed with default parameters using both the complete manatee IgA sequence and a trimmed region comprising constant

domains CH1, CH2, and CH3. For visualization, we created trees using a unique reference sequence per species, and the NJ trees were edited using iTOL [17] to improve visualization.

## Structural comparison

We searched for manatee's IgA in AlphaFold [18,19], and access codes were used for a search in Foldseek [20]. This search aimed to identify proteins exhibiting a high degree of structural homology with IgA. Subsequently, graphical analysis was employed to compare similar domains between the identified proteins and the manatee's IgA. The assessment involved Template Modelling score (TM-score) and Root Mean Square Deviation (RMSD) values to ascertain the most comparable folds.

## Prediction of linear B-cell epitopes

We predicted linear B-cell epitopes on all available manatee IgA sequences using at least 10 different algorithms across three servers: BcePred [21], IEDB (Immune Epitope Database and Analysis Resource) [22], and ABCpred [23,24]. BcePred evaluates seven physicochemical properties of each amino acid for epitope prediction, including hydrophilicity [25], flexibility [26], accessibility [27], turns [28], surface exposure [29], polarity [30], and antigenic propensity [31]. Each amino acid is assigned values ranging from -3 to +3, with thresholds varying for each property. We manually selected amino acids based on whether they passed specific default thresholds for each property, constructing our consensus epitope for every IgA manatee sequence.

On the other hand, IEDB utilizes experimentally derived B-cell epitope data from humans and other species across different medical contexts. The server employs seven algorithms for epitope prediction; however, we specifically utilized Bepipred-1.0 Linear Epitope Prediction [32] and BepiPred-2.0: Sequential B-Cell Epitope Prediction [33], with threshold values ranging from -0.2 to +1.3 and 0 to +1, respectively. Default threshold values were applied.

ABCpred utilizes artificial neural networks trained on a database containing 700 experimentally tested unique epitopes. The platform employs a threshold ranging from +0.1 to +1, with the default value set at 0.51. To address the challenge of the high quantity and overlap of predicted epitopes, we selected a single epitope from each region of the sequence, guided by previously predicted epitopes.

Consensus epitopes were determined for each algorithm when the same amino acid in the predicted epitope was present in three or more sequences. The most conserved fragments were then selected as the consensus epitopes. Finally, a consensus epitope among all the algorithms was determined based on the parameters previously mentioned.

## Prediction of conformational B-cell epitopes

Conformational epitopes consist of discontinuous amino acids in close proximity after protein folding [34]. The DiscoTope 3.0 server [35] was used for these predictions. DiscoTope 3.0 is a server that predicts discontinuous B-cell epitopes from their three-dimensional structure, using reverse folding methods for epitope determination. AlphaFold IDs were used as the source of structure, and a higher confidence threshold ($>1.5$ calibrated score) was used. All amino acids that passed the detection threshold were considered conformational epitopes. Conformational and linear epitopes were located in the predicted manatee structure retrieved from AlphaFold (AF-A0A1S6EEL0-F1) using PyMol [36].

## Conservation, identity, and similarity assessment

A conservation score was established to evaluate the reliability of the epitope predictions. Initially, an intra-algorithm analysis was conducted to obtain a score that reflected if: 1) Is the epitope present in how many out of the five available manatees IgA sequences? 2) How well conserved is every amino acid position in the predicted epitopes? 3) What is the average score per algorithm of the conserved epitope? 4) Using an interquartile normalization strategy, in which interquartile range localizes every average epitope score? (see supplementary methods). Subsequently, an inter-algorithm analysis was carried out to compute our final conservation score. Briefly, we assessed if: 1) How many algorithms predicted the same or similar epitopes? and 2) How well conserved is every amino acid position in the predicted epitopes? All these results were standardized on a scale from 0 to 1, penalizing whenever a mismatch amino acid was encountered, or the epitope was not predicted. We finally computed an overall conservation index for every epitope by averaging every individual score obtained.

To assess the performance of our conservation index, we employed two methods to confirm the presence of the predicted epitope on human IgA sequences. Firstly, we conducted BLATS® - Global Alignment (Needleman-Wunsch) analysis to determine if the epitope was present. Subsequently, we trimmed the specific target sequence to obtain an identity value for each predicted epitope (as query). Secondly, we conducted a similarity comparison by searching the IEDB database for previously identified epitopes on human IgA (IGHA1 [P01876] and IGHA2 [P01877]). These epitopes have been experimentally identified in the context of infectious diseases, allergies, autoimmunity, and transplantation. After searching, we manually assessed conservation by computing a score that considers the number of shared amino acids and the total number of epitopes that shared any amino acid with the predicted manatee epitope (see supplementary methods).

## Ethical considerations

We used publicly available sequences; therefore, no ethical considerations need to be taken into account.

## Results

### Sequence retrieval, processing, and alignment

We identified five partial IgA sequences of The Florida manatee *(Trichechus manatus latirostris)* with varying lengths (389–466 aa) on the NCBI servers (as of February 1st, 2024). These sequences were obtained and uploaded by Breaux et al. in 2017 through the sequencing of IgH RACE PCR products from a single individual [14,37]. In Fig 1, the tree displays the sequences obtained from BLAST hits at Uniprot. We used a trimmed sequence representing the constant heavy regions from the longest available IgA protein sequence (A0A1S6EEL0) as the query. Accession numbers, query cover, and identity percentage for the selected results are provided in S1 and S2 Tables in S1 File. Additionally, S1 Fig presents similar trees using the longest complete sequence at Uniprot (S1A Fig) and NCBI (S1B Fig), along with the trimmed sequence at NCBI (S1C Fig) as queries.

The generated trees offer insights into the similarities in amino acid sequences among the analyzed species through BLASTp and alignment processes. We observed that species tended to cluster based on their taxonomic order classification. However, the relationships between orders are organized differently compared to the established phylogeny. For instance, there is an unexpected proximity between the orders Sirenia and Primates, and the Rodentia order is more separated than anticipated (see Figs 1 and S1). Nevertheless, there is a notable identity

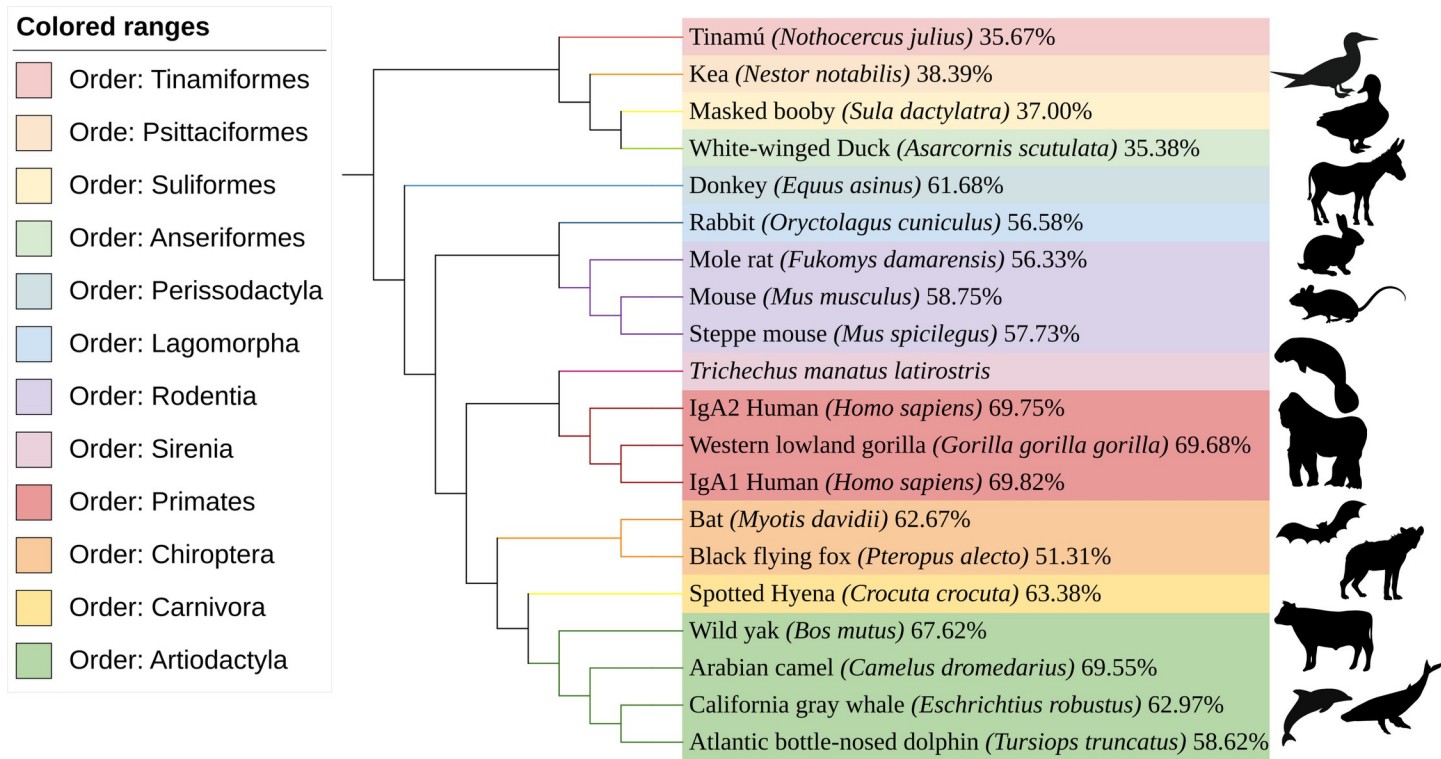

**Fig 1. UniProt neighbor joining tree.** Neighbor-joining tree constructed using the constant heavy regions of the longest manatee IgA as a query at UniProt. The tree reveals a notable proximity between the orders Sirenia and Primates, with identities surpassing 69%.

percentage between the manatee's IgA sequences and those from Primates, suggesting a remarkable similarity in the IgA sequences between these two orders.

## Structural comparison

The structure prediction for the longest IgA sequence from the manatee was retrieved from Alpha Fold (AF-A0A1S6EEL0-F1). The structure exhibits mixed very high (>90) and high (90 > 70) model confidence scores for the VH, CH1, CH2, and CH3 domains. Conversely, the interdomain regions have low (70 > 50) and very low (<50) model confidence scores. This information is crucial for the upcoming structural comparisons we present. The predicted aligned error (PAE) reveals a similar pattern, with lower errors for the domains and higher errors for the interdomain regions.

Using this model, we conducted a Foldseek search, and the majority of the hits were for human structures. Subsequently, we selected TM-scores, RMSD values, and target lengths from six different species to illustrate their overlap with manatee IgA (refer to Table 1). The TM score, ranging from 0 to 1, indicates that proteins with a score above 0.5 are mostly in the same fold as the query [38]. In this case, TM scores ranged from 0.61440 to 0.71339, signifying a high structural similarity with manatee IgA. Specifically, IgA1 and IgA2 from humans exhibited TM scores of 0.68350 and 0.64365, respectively. For RMSD values, which provide information about structural comparison between two proteins, a value $\leq 2$ Å is considered fairly good [39]. Although none of the hits are $\leq 2$ Å, the lowest values were observed for Homo sapiens IgA1 and IgA2. Based on these values, human IgA appears to have the most similar

**Table 1. Selected Foldseek results for six different species.**

| Species name | Seq. ID | TM score | RMSD | Target length |
|---|---|---|---|---|
| *Homo sapiens*[α] | 66.3 | 0.68350 | 8.15 | 353 |
| *Homo sapiens*[β] | 68.6 | 0.64365 | 9.82 | 340 |
| *Mus musculus* | 57.6 | 0.63717 | 14.67 | 344 |
| *Gorilla gorilla gorilla* | 66.3 | 0.62753 | 10.34 | 353 |
| *Equus asinus* | 58.3 | 0.61440 | 10.93 | 346 |
| *Oryctolagus cuniculus* | 54.8 | 0.71339 | 10.91 | 229 |
| *Tursiops truncatus* | 59.1 | 0.67183 | 18.25 | 405 |

TM: Template modelling; RMSD: Root mean square deviation.

α: IgA1; β: IgA2.

structure to manatee IgA. This conclusion is drawn from the highest TM-scores, lowest RMSD values, and query cover.

When considering the overlap, as depicted in Fig 2, we found that: 1) The VH domain is absent from the human structure (yellow) in both cases; 2) the CH1 domain does not overlap with human IgAs, primarily due to its positioning relative to other domains, and this might be related to the low confidence in the structural prediction for the interdomain region between CH1 and CH2; however, a detailed observation allows us to note a close similarity for the CH1 domain itself; and 3) Notably, the CH2 and CH3 domains demonstrate the highest overlap with the structure of both human IgA1 and IgA2.

## Prediction of linear B-cell epitopes

Thirteen consensus epitopes were identified based on the predictions of the 10 algorithms used. These epitopes were distributed across different domains, with two in the VH domain, four in the CH1 domain, three in the CH2 domain, and four in the CH3 domain (Table 2 and Fig 3). The minimum number of algorithms detecting each epitope was arbitrarily set as three,

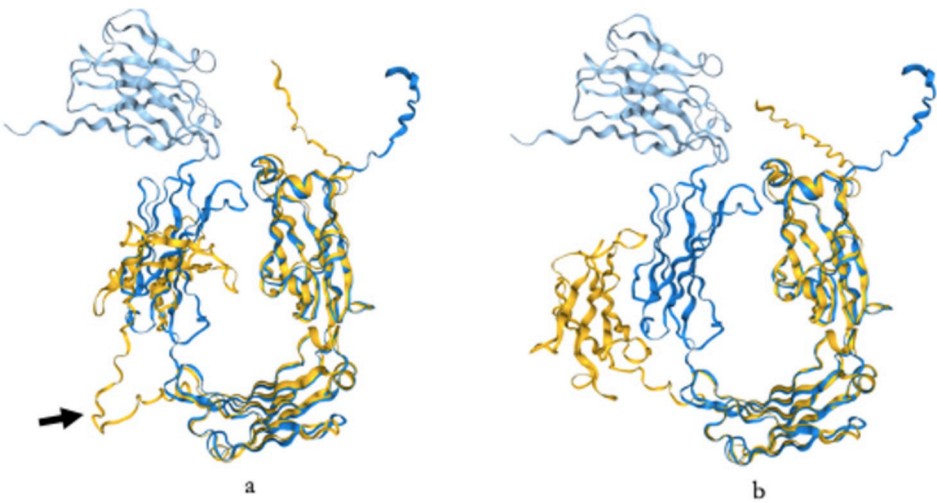

**Fig 2. Stuctural overlap between manatee and human IgA.** Structural overlaps obtained with Foldseek. (a.) Manatee IgA (Blue) *vs*. Human IgA1 (Yellow). (b.) Manatee IgA (Blue) *vs*. Human IgA2. CH2 and CH3 exhibit the highest structural similarity. CH1 is not as close as expected and this might be due to the interdomain region (arrow) which is longer in humans and might be affecting the overlapping. However, the CH1 itself is very similar.

**Table 2. Conservation, identity and similarity of predicted epitopes along manatee IgAs.**

| Epitope | Domain | Conservation | Identity | | Similarity w IgA1 | | | Similarity with IgA2 | | |
|---|---|---|---|---|---|---|---|---|---|---|
| | | | IgA1 | IgA2 | M | R | # | M | R | # |
| DTSKS | VH | 0.7330 | - | - | - | - | - | - | - | - |
| VNSEDT | VH | 0.7438 | - | - | - | - | - | - | - | - |
| LVTVSSEPETSPRVFP | CH1 | 0.7617 | 0,44 | 0,44 | 4/16 | 4/16 | 2 | - | - | - |
| NHSGENVTV | CH1 | 0.7524 | 0,56 | 0,56 | - | - | - | - | - | - |
| DQCPDN | CH1 | 0.7575 | 0,33 | 0,67 | 2/6 | 2/6 | 1 | - | - | - |
| HNSSSQEAKVP | CH1 | 0.7141 | 0,36 | 0,45 | 2/11 | 1/11–4/11 | 27 | 1/11 | 1/11 | 1 |
| APERDS | CH2 | 0.7515 | 0,67 | 0,67 | 4/6 | 3/6-4/6 | 4 | 4/6 | 4/6 | 4 |
| EPWKSGNK | CH2 | 0.7477 | 0,71 | 0,43 | 2/8 | 2/8 | 4 | - | - | - |
| GTQSATISKNSGN | CH2 | 0.7606 | 0,57 | 0,31 | 3/13 | 3/13 | 10 | 3/13 | 3/13 | 1 |
| HLLPPPAEEL | CH3 | 0.7086 | 0,90 | 0,90 | 9/10 | 1/10-9/10 | 9 | 9/10 | 6/10-9/10 | 10 |
| QLPQNN | CH3 | 0.7722 | 0,50 | 0,50 | 2/6 | 2/6 | 12 | 2/6 | 1/6-2/6 | 18 |
| PRQEPG | CH3 | 0.6702 | 0,67 | 0,67 | 4/6 | 1/6-4/6 | 39 | 1/6 | 1/6 | 1 |
| QTWKWGD | CH3 | 0.7269 | 0,57 | 0,42 | 2/7 | 1/7-4/7 | 56 | 2/7 | 2/7 | 1 |

M: Mode of shared amino acids with epitopes reported at IEDB; R: Range of shared amino acids with epitopes reported at IEDB.

#: Number of epitopes reported at IEDB with which existed at least 1 shared amino acid.

so that we show those epitopes predicted by three or more algorithms. The percentage of identity between the identified epitopes and the human IgA1 and IgA2 sequences ranged from 31% to 90%, with the consensus epitope "HLLPPPAEEL" exhibiting the highest identity (90%). The complete set of epitope predictions for each sequence and algorithm, along with their respective scores, can be found in S3 Table in S1 File. The spatial distribution of consensus epitopes within the protein structure (Fig 3) indicates that these epitopes are predominantly present in accessible regions of the protein.

## Prediction of conformational B-cell epitopes

Amino acids that might be part of conformational epitopes were predicted with DISCOTOPE 3.0 [35]. Those amino acids that matched with the previously predicted linear epitope are highlighted in Table 2 and positioned within the protein structure in Fig 3 as blue sectors. All predictions of the conformational B-Cell epitopes are shown in S2 Fig.

## Conservation, identity, and similarity

Based on the previous results, we evaluated the conservation of the predicted epitopes across the different algorithms and within the sequences of manatee's IgA. This conservation analysis generated scores and was conducted for the 13 predicted epitopes as outlined in the methodology. Additionally, we performed a shared amino acid analysis with confirmed existing epitopes reported on the IEDB website, as exemplified in S4 Table in S1 File. In total, 254 epitopes were identified in IEDB for IGHA1 and 99 for IGHA2. Our predicted epitopes matched with a total of 164 and 36 IEDB epitopes for IGHA1 and IGHA2, respectively. Table 2 presents the predicted epitopes, their domain locations, conservation scores, sequence identity values, and details on how many epitopes our predictions matched with at IEDB, including the mode, range, and total. Despite the epitope "HLLPPPAEEL" not displaying the best conservation score, it exhibited excellent identity and similarity values.

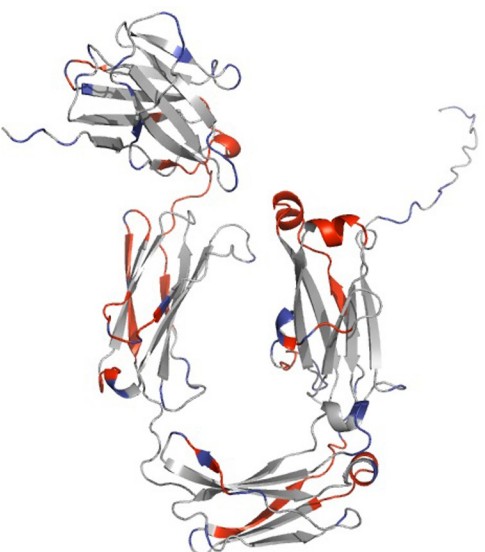 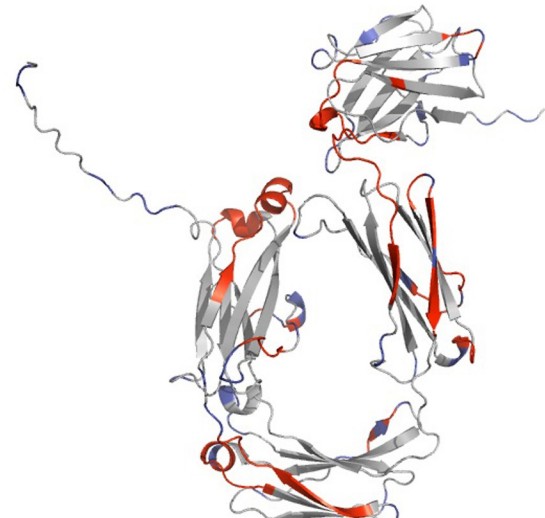

**Fig 3. Linear and conformational epitope prediction.** Predicted linear (red) and conformational epitopes (blue) within the longest manatee IgA sequence. Panel a show a front view and panel b shows a rear view.

## Discussion

In this study, we conducted an immuno-informatics analysis to compare IgA sequences from manatees with those from other species, with the aim of exploring epitope similarity and sharing. Human IgA emerged as the sequence with the highest similarity values compared to manatee IgA, as supported by our findings across sequence, 3D-structure, and B-cell epitope comparisons. To our knowledge, this study represents the first attempt to identify similar sequences to manatee IgA using all three parameters. Previous studies have reported similar findings, with human IgA showing the highest identity values compared to sequences from other species, including dolphin, cow, and dog [14]. However, structural, and B-cell epitope comparisons with manatee IgA sequences have not been conducted previously.

The sequence comparison conducted in this study revealed that primate sequences exhibit the highest identity percentages with the manatee sequences, aligning with previous research showing higher identity percentages for human IgA2 followed by human IgA1[14]. Interestingly, in terms of tree topology, the order Primates showed closer proximity to Sirenia than other orders phylogenetically closer to Primates, such as Rodentia or Lagomorpha. This differs from a previous comparison of IgA in terrestrial and marine mammals, which did not demonstrate a similar tree topology between manatee and primate IgA [40]. It's worth noting that we could not replicate those results, and a notable difference was that while our trees were constructed using protein sequences of in vitro transcribed data from conserved domains exclusively, the previous study utilized IGHC gene sequences.

The RMSD, TM score, and query cover values from the structural comparison indicate that human IgA, specifically IgA1, most closely resembles the predicted fold of manatee IgA. The overlap is more pronounced in CH2 and CH3 regions, suggesting potential cross-reactivity. There are no previous studies comparing these two structures or in general human and manatee proteins to make a comparison identifying if the obtained values represent a good fold. However, the association of a similar 3D fold with a higher probability of cross-reactivity has been documented in previous studies [41,42].

The deployment of 10 different algorithms for the B-cell epitope prediction of the manatee IgA sequence offered a diverse range of key characteristics crucial for epitope definition, including hydrophilicity, accessibility, and mobility. These characteristics have been reported as pivotal factors in determining epitope positions [43]. The conservation score generated revealed that the consensus epitopes identified within and across algorithms exhibited a high level of conservation, indicating that the region of the sequence is likely a real epitope. Furthermore, it is well-established that conformational epitopes constitute the majority of epitopes found in proteins, and antibody recognition tends to be stronger compared to linear epitopes [43]. Therefore, the presence of predicted conformational epitopes within the linear epitopes provides more certainty to the nature of the fragment as an epitope.

The comparison between the predicted consensus epitopes and the human IgA sequences revealed that the selected fragments indeed appeared in the human sequence, albeit with varying identity values. Furthermore, when compared with experimentally proven epitopes in humans, a substantial number of hits were obtained. This suggests not only the presence of sequence fragments in both sequences with high identity values but also their recognition as epitopes in human IgA, with documented antibody binding.

Human IgA's similarity to manatee IgA across the three studied parameters suggests the possibility of cross-reactivity with human anti-IgA antibodies, especially polyclonal antibodies. Human reagents have successfully identified IgA in the Asian elephant (*Elephas maximus*), a species closely related to the manatee, in various bodily fluids, including feces, saliva, urine, and serum [44]. Furthermore, anti-human antibodies have been employed to detect the acute-phase protein serum amyloid A in manatees [45]. However, it's essential to note that when using non-species-specific reagents to identify other immune molecules such as cytokines, CRP, and haptoglobin in manatees, the results have not been equally successful [46,47]. In these studies, monoclonal antibodies were used, so this could be a reason for the lack of cross-reactivity given that targets might not share the same exact epitope sequence. For future functional studies, it is crucial to consider the nature of the reagents used. Employing polyclonal antibodies, which recognize multiple epitopes, could enhance the probability of detecting some of the epitopes that showed high identity values with human IgA.

The utilization of computational biology tools represents a significant advancement in the future exploration of immunology in non-human animals, addressing persistent challenges associated with the scarcity of species-specific reagents. The model proposed in this study offers a valuable opportunity to enhance the scope of studies in these species by providing insights into which reagents are more likely to be effective in each case.

In the context of conservation implications for manatees, the study of their immune system contributes to a deeper understanding of their relationship with the evolving environment and the impacts of prevalent health threats faced by these animals. Furthermore, it opens opportunities for new research on the repercussions of the lack of breastfeeding in orphan calves, aiming to develop innovative strategies for safeguarding these individuals, thereby increasing their survival rates and overall quality of life. Functional analysis needs to be conducted using the suggested species-specific reagents in order to prove the cross-reactivity and make advancements in the identification and quantification of manatee IgA. Such studies are being performed by our group as we write this manuscript.

It's crucial to emphasize that the entire analysis was conducted using sequences obtained from blood samples. In this context, IgA exists as a monomer without the secretory component and the joining chain (J-chain). In secretory IgA (sIgA), the secretory component (SC) is bound to one face of the dimer and has contact with the constant region of both Ig chains and the J-chain. The primary function of SC is to protect sIgA from proteolytic cleavage [48]. The J-chain binds to specific sites in the constant region of both Ig chains, particularly to the tail

piece of the protein, and its primary role is to assemble the dimer and facilitate the transport of Ig across mucosal epithelia [49]. The presence of SC and J-chain may influence the recognition by an antibody targeted to the constant region, introducing potential allosteric restrictions that can hinder antigen-antibody interaction.

In studies predicting cross-reactivity in the context of allergic reactions, it is well-established that the three components analyzed in the present study (sequence, structure, epitope similarity) are fundamental for creating an accurate prediction [41,42]. However, in these types of studies, the results are typically compared with a dataset of known allergens, providing a higher confidence in the predictions of cross-reaction. Currently, there are no established models for predicting cross-reactivity between species for the identification of potential reagents for successful functional experiments. Additionally, datasets and algorithms for predicting cross-reactivity, especially B-cell epitope prediction, were primarily developed for research in the human medical field, particularly in the context of vaccine development. This means that the parameters in which the predictions are made might not be as reliable for other species, even though they are based on the global principles of epitope determination. Therefore, the confidence of the prediction needs to be validated *in vitro*, and other factors specific to the species in question might need to be considered in order to ensure the accuracy of the predictions.

In conclusion, our study represents a pivotal step forward in elucidating the immunological landscape of manatees, offering valuable insights into their immune system and potential cross-reactivity with human IgA. The utilization of computational biology tools holds immense promise in expanding our understanding of immunology in non-human animals, addressing persistent challenges associated with the scarcity of species-specific reagents. Moving forward, future studies should focus on functional analysis using suggested species-specific reagents to validate cross-reactivity and advance our understanding of manatee IgA. Such endeavors are critical not only for advancing our knowledge but also for informing conservation efforts and safeguarding these remarkable creatures for generations to come.

## Supporting information

**S1 Fig. A:** Tree generated from the complete IgA sequences using the hits from Uniprot. **B:** Tree of the complete IgA sequence using the hits from NCBI. **C:** Tree of the constant regions of IgA (CH1, CH2, CH3) using the hits from NCBI.
(TIF)

**S2 Fig. All conformational epitopes located in the 5 manatee IgA sequences.** X represent amino acids that passes the detection threshold during the prediction.
(TIF)

**S1 File. S1 Table. NCBI and UniProt hits of complete IgA**. Accession number, species name, sequence name and identity percentage for each hit. **S2 Table. NCBI and UniProt hits of constant regions (CH1, CH2, CH3) of IgA.** Accession number, species name, sequence name and identity percentage for each hit. **S3 Table. Consensus Linear B-cell Epitopes and Scores. S4 Table. Example of predicted manatee epitope and confirmed human epitope comparison.** ID of the epitope sequence on the IEDB data base; Shared amino acids between the predicted epitope on the manatee IgA sequence with the confirmed epitopes in the IDEB database; Position over the reference sequence of human IgA1.
(PDF)

**S2 File. This file details the conservation score established to evaluate the reliability of the epitope predictions.** The analysis includes: 1) The presence of the epitope in the five available

manatee IgA sequences; 2) The conservation level of each amino acid position within the predicted epitopes; 3) The average conservation score per algorithm for the conserved epitopes; and 4) An interquartile normalization strategy, showing the localization of each average epitope score within the interquartile range.
(DOCX)

## Acknowledgments

During the preparation of this work the author used ChatGPT-3.5 in order to review the syntax and grammar of the document. After using this tool/service, the author reviewed and edited the content as needed and takes full responsibility for the content of the publication.

## Author Contributions

**Conceptualization:** Mariapaula Díaz-Yayguaje, Susana Caballero-Gaitan, Augusto Valderrama-Aguirre.

**Data curation:** Mariapaula Díaz-Yayguaje, Augusto Valderrama-Aguirre.

**Formal analysis:** Mariapaula Díaz-Yayguaje, Augusto Valderrama-Aguirre.

**Funding acquisition:** Augusto Valderrama-Aguirre.

**Investigation:** Mariapaula Díaz-Yayguaje, Susana Caballero-Gaitan, Augusto Valderrama-Aguirre.

**Methodology:** Mariapaula Díaz-Yayguaje, Augusto Valderrama-Aguirre.

**Project administration:** Augusto Valderrama-Aguirre.

**Resources:** Susana Caballero-Gaitan, Augusto Valderrama-Aguirre.

**Supervision:** Augusto Valderrama-Aguirre.

**Validation:** Mariapaula Díaz-Yayguaje, Augusto Valderrama-Aguirre.

**Visualization:** Mariapaula Díaz-Yayguaje, Augusto Valderrama-Aguirre.

**Writing – original draft:** Mariapaula Díaz-Yayguaje, Augusto Valderrama-Aguirre.

**Writing – review & editing:** Mariapaula Díaz-Yayguaje, Susana Caballero-Gaitan, Augusto Valderrama-Aguirre.

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
