## [Decision Letter · Decision Letter 0]

5 Jun 2024

PONE-D-24-17846Unlocking epitope similarity: a comparative analysis of manatee (Trichechus manatus) IgA and human IgA through an immuno-informatics approach

PLOS ONE

Dear Dr.  Valderrama-Aguirre,

Thank you for submitting your manuscript to PLOS ONE. After careful consideration, we feel that it has merit but does not fully meet PLOS ONE’s publication criteria as it currently stands. Therefore, we invite you to submit a revised version of the manuscript that addresses the points raised during the review process.

We look forward to receiving your revised manuscript.

Kind regards,

Karolina Goździewska-Harłajczuk

Academic Editor

PLOS ONE

Journal Requirements:

   "This work was supported by a FAPA project (PVI0122029) from Universidad de Los Andes, granted to A.V.A."  

Additional Editor Comments:

In general the subject of the manuscript is worthy of publication, however there are some major points which need an improvement. The Reviewers suggest the revision of your paper. Please make all the necessary correction of your manuscript.

Reviewers' comments:

Reviewer's Responses to Questions

**Comments to the Author**

1. Is the manuscript technically sound, and do the data support the conclusions?

Reviewer #1: Partly

Reviewer #2: Yes

2. Has the statistical analysis been performed appropriately and rigorously? 

Reviewer #1: No

Reviewer #2: Yes

3. Have the authors made all data underlying the findings in their manuscript fully available?

Reviewer #1: Yes

Reviewer #2: Yes

4. Is the manuscript presented in an intelligible fashion and written in standard English?

Reviewer #1: No

Reviewer #2: Yes

5. Review Comments to the Author

Reviewer #1: -Title: include the vernacular name;

-Short title: inclued "manatee" or vernacular name;

-If you use the vernacular name of the subspecies "Caribbean manatee", provide the scientific name of the subspecies (Trichechus manatus manatus). If you are going to use only Trichechus manatus, use the vernacular name American manatee or just manatee;

-the vernacular name of the subspecies should be "Antillean manatee" or "Weste Indian manatee" or "Greater Caribbean manatee", but the "Geater" was not included in the name;

- Abstract is subjective. Need to improve writing;

- Reference "1" is a regional action plan. Use a reference with greater global information to inform distribution;

- material and methods need to come before results;

-Inform whether the milk supplied to the manatee is with or without lactose;

-I believe that reference 7 is not adequate to talk about the nutritional issues of manatees in captivity;

-I strongly recommend reviewing references for the information they provide before citing them;

- The text does not make it clear whether the information on conservation comes from the study site or the entire area where the species occurs, especially with regard to food and the problems caused in captivity;

- It is necessary to check the "Submission Guidelines" item to adapt the manuscript to the journal's standards;

-vernacular and scientific name formats need to be standardized throughout the text. A rigorous review is necessary. the vernacular name must be outside the parenthesis and the scientific name inside;

- I didn't understand why from the "Supplementary Methods" file, this information can be described in the text;

- The text presents important results, but it needs to undergo text organization, adoption of standards and adaptation of names, before a more appropriate evaluation.

-I request an initial adaptation and organization of the text, so that it can be evaluated again

Reviewer #2: Authors performed an inmune-informatic study to compare manate IgA with that of other species, as a base knowledge to design further studies on manate IgA, specially in manatee breast-milk. Authors used available sequences and use computational algorithms to compare different aspects of IgA from other species.

I think it is a well designed study presented in a clear manuscript. My only coments are:

a) Considere to use "American manatee" for the species, as a recent exercise from many specialists propose it in Mignucci et al. [in press Cariben Naturalist].

b) I think, author used the only species which have available sequences of IgA in public databases, but is not clear in the methods if this is true or if they took some species. In this sense, I wonder if there is IgA data on other afrotherians, like elephants or among other manatee species or subspecies.

c) In discussion, several times authors repeat the statement about the finding of more similarity with human IgA, I found it some kind redundant.

6. PLOS authors have the option to publish the peer review history of their article (what does this mean?). If published, this will include your full peer review and any attached files.

Reviewer #1: **Yes: **Fernanda Loffler Niemeyer Attademo

Reviewer #2: No

---

## [Author Response · Author response to Decision Letter 0]

3 Jul 2024

Bogotá DC, Colombia; June 28, 2024

Karolina Goździewska-Harłajczuk

Academic Editor PLOS ONE

Re: PONE-D-24-17846 Response to Reviewers

Respected Academic Editor and Reviewers,

Thank you for the opportunity to revise and resubmit our manuscript titled "Unlocking epitope similarity: a comparative analysis of the American manatee (Trichechus manatus) IgA and human IgA through an immuno-informatics approach." Reviewer’s insightful comments and suggestions have significantly enhanced the quality of our manuscript. We have carefully considered each of your comments, and the changes made are highlighted throughout the manuscript with track changes. Below, we provide detailed responses to each of the suggestions and comments.

Response: We have thoroughly revised the manuscript to ensure it meets PLOS ONE's style requirements, including adhering to the appropriate file naming conventions.

2. Please note that PLOS ONE has specific guidelines on code sharing for submissions in which author-generated code underpins the findings in the manuscript. In these cases, we expect all author-generated code to be made available without restrictions upon publication of the work.

Response: Since our research was conducted relying upon freely available software and did not involve generating author-specific code, there are no codes underpinning our findings to share.

Response: We have removed all funding-related text from the manuscript as per your guidelines. This information will now only be included in the Funding Statement section of the online submission form.

Reviewer #1 comments

1. Title: include the vernacular name.

Response: We have included the vernacular name "American manatee" in the title as recommended.

2. Short title: include "manatee" or vernacular name

Response: We have included the vernacular name "American manatee" in the short title as "American manatee IgA cross reactivity."

3. If you use the vernacular name of the subspecies "Caribbean manatee", provide the scientific name of the subspecies (Trichechus manatus manatus). If you are going to use only Trichechus manatus, use the vernacular name American manatee or just manatee. The vernacular name of the subspecies should be "Antillean manatee" or "Weste Indian manatee" or "Greater Caribbean manatee", but the "Geater" was not included in the name.

Response: We have opted to consistently use "American manatee" (Trichechus manatus) as both the vernacular and scientific name throughout the text. While "American manatee" is used in the titles and initial mentions in the abstract and introduction, we use "manatee" subsequently in the text to maintain clarity and consistency.

4. Abstract is subjective. Need to improve writing.

Response: We have revised the abstract to enhance clarity and objectivity, ensuring it effectively communicates the study's objectives and results. Additionally, the text has been reviewed by both AI ChatGPT and an anglophone style revisor for further refinement.

5. Reference "1" is a regional action plan. Use a reference with greater global information to inform distribution.

Response: We have considered your feedback and replaced the reference with a source that offers a broader global perspective on the species' distribution.

6. Material and methods need to come before results.

Response: We have restructured the manuscript accordingly, with the Methods section now preceding the Results.

7. Inform whether the milk supplied to the manatee is with or without lactose.

Response: We have addressed the presence of lactose in milk substitutes in the introduction, detailing its concentrations and implications for manatee health.

8. I believe that reference 7 is not adequate to talk about the nutritional issues of manatees in captivity.

Response: We have reviewed the references and selected more appropriate sources to discuss the nutritional issues of manatees in captivity. The necessary changes have been made accordingly.

9. I strongly recommend reviewing references for the information they provide before citing them. 

Response: We have thoroughly reviewed the references with two independent reviewers and updated some of them based on the information provided in each article.

10. The text does not make it clear whether the information on conservation comes from the study site or the entire area where the species occurs, especially with regard to food and the problems caused in captivity.

Response: We have revised the manuscript to clarify that the conservation information pertains to the entire range where the species is found, as supported by the references cited.

11. It is necessary to check the "Submission Guidelines" item to adapt the manuscript to the journal's standards.

Response: Following the reviewer's 1 suggestion, we have carefully examined the 'Submission Guidelines' and made necessary adjustments to ensure that the manuscript conforms to the journal's standards.

12. Vernacular and scientific name formats need to be standardized throughout the text. A rigorous review is necessary. the vernacular name must be outside the parenthesis and the scientific name inside.

Response: We have standardized the format of vernacular and scientific names throughout the manuscript as per your suggestion, with the vernacular name outside parentheses and the scientific name inside.

13. I didn't understand why from the "Supplementary Methods" file, this information can be described in the text.

Response: Thank you for your observation. We considered that including the detailed information from the "Supplementary Methods" file could potentially overwhelm the reader's attention from the main Methods section. However, we aimed to ensure comprehensive coverage of the data analysis process.

14. The text presents important results, but it needs to undergo text organization, adoption of standards and adaptation of names, before a more appropriate evaluation. I request an initial adaptation and organization of the text, so that it can be evaluated again. 

Response: Thank you for your comprehensive evaluation of our manuscript and for your valuable comments. We appreciate the suggestions for improving text organization, adopting standards, and ensuring name consistency. These changes have been meticulously implemented to align with journal requirements. We look forward to your reassessment of the revised manuscript.

Reviewer #2 comments

1. “Authors performed an inmune-informatic study to compare manate IgA with that of other species, as a base knowledge to design further studies on manate IgA, specially in manatee breast-milk. Authors used available sequences and use computational algorithms to compare different aspects of IgA from other species. I think it is a well designed study presented in a clear manuscript.” 

Response: Thank you sincerely for your positive assessment of our study and manuscript.

2. Considere to use "American manatee" for the species, as a recent exercise from many specialists propose it in Mignucci et al. [in press Cariben Naturalist].

Response: Thank you for your suggestion regarding the vernacular name. We have opted to consistently use "American manatee" throughout the article, as recommended by specialists (Mignucci et al., in press, Caribbean Naturalist).

3. I think, author used the only species which have available sequences of IgA in public databases, but is not clear in the methods if this is true or if they took some species. In this sense, I wonder if there is IgA data on other afrotherians, like elephants or among other manatee species or subspecies.

Response: Thank you for raising this point. To address the study's focus on homology comparison, we specifically limited our analysis to the top 100 results in BLASTp, as detailed in the manuscript. Furthermore, we confirmed that IgA sequences from other Afrotheria species or additional manatee species are currently unavailable and were therefore not included in our study. These clarifications have been incorporated into the revised manuscript.

4. In discussion, several times authors repeat the statement about the finding of more similarity with human IgA, I found it some kind redundant.

Response: Thank you for highlighting this issue. We appreciate your feedback and have revised the discussion section accordingly to eliminate redundancy and improve clarity.

We sincerely appreciate the insightful feedback provided by the reviewers. Their comments have been invaluable in refining our manuscript to meet the rigorous standards of PLOS ONE. We have diligently addressed each point raised, including standardizing the vernacular and scientific names, clarifying methods and results sections, and ensuring adherence to submission guidelines. These revisions have strengthened the clarity, organization, and scientific rigor of our manuscript. We are confident that these enhancements significantly improve the manuscript and look forward to the reviewers' reassessment. 

Thank you once again for the opportunity to improve our work and for your consideration. We hope that our manuscript will be given careful consideration for publication in PLOS ONE.

Sincerely yours,

AUGUSTO VALDERRAMA-AGUIRRE, M.SC. PH.D.

Assistant Professor, Department of Biological Sciences

Universidad de Los Andes, Bogotá DC, Colombia

---

## [Decision Letter · Decision Letter 1]

24 Jul 2024

Unlocking epitope similarity: a comparative analysis of the American manatee (Trichechus manatus) IgA and human IgA through an immuno-informatics approach

PONE-D-24-17846R1

Dear Dr. Valderrama-Aguirre,

We’re pleased to inform you that your manuscript has been judged scientifically suitable for publication and will be formally accepted for publication once it meets all outstanding technical requirements.

Kind regards,

Karolina Goździewska-Harłajczuk

Academic Editor

PLOS ONE

Additional Editor Comments (optional):

All comments have been addressed. The manuscript can be accept in its current form.

Reviewers' comments:

Reviewer's Responses to Questions

**Comments to the Author**

1. If the authors have adequately addressed your comments raised in a previous round of review and you feel that this manuscript is now acceptable for publication, you may indicate that here to bypass the “Comments to the Author” section, enter your conflict of interest statement in the “Confidential to Editor” section, and submit your "Accept" recommendation.

Reviewer #1: All comments have been addressed

Reviewer #2: All comments have been addressed

2. Is the manuscript technically sound, and do the data support the conclusions?

Reviewer #1: (No Response)

Reviewer #2: Yes

3. Has the statistical analysis been performed appropriately and rigorously? 

Reviewer #1: (No Response)

Reviewer #2: Yes

4. Have the authors made all data underlying the findings in their manuscript fully available?

Reviewer #1: Yes

Reviewer #2: Yes

5. Is the manuscript presented in an intelligible fashion and written in standard English?

Reviewer #1: Yes

Reviewer #2: Yes

6. Review Comments to the Author

Reviewer #1: (No Response)

Reviewer #2: Authors fully adressed the reviewrs observations and comments. This is a valuable preliinary study on manatees inmunology that surely will be followed by further Works on this item that Will help on management of the species, specially on orphan cañves´s rehabilitation and care.

7. PLOS authors have the option to publish the peer review history of their article (what does this mean?). If published, this will include your full peer review and any attached files.

Reviewer #1: **Yes: **Fernanda Loffler Niemeyer Attademo

Reviewer #2: No

---

## [Editor Report · Acceptance letter]

2 Aug 2024

PONE-D-24-17846R1 

PLOS ONE

Dear Dr. Valderrama-Aguirre, 

I'm pleased to inform you that your manuscript has been deemed suitable for publication in PLOS ONE. Congratulations! Your manuscript is now being handed over to our production team.

Kind regards, 

on behalf of

Dr. Karolina Goździewska-Harłajczuk 

Academic Editor

PLOS ONE